# Influence of the Thickness of the Seasonally Thawed Layer of Permafrost in the Eastern Siberia Catchments on the Content of Organic Matter in River Waters

Olga I. Gabysheva [1], Viktor A. Gabyshev [2], Sophia Barinova [3,*], Irina A. Yakshina [2] and Innokentiy S. Pavlov [4]

[1]  Institute for Biological Problems of Cryolithozone Siberian Branch of Russian Academy of Science (IBPC SD RAS), 677980 Yakutsk, Russia; g89248693006@yandex.ru

[2]  The State Nature Reserve Ustlensky, 678400 Tiksi, Russia; v.a.gabyshev@yandex.ru (V.A.G.); i_yakshina@rambler.ru (I.A.Y.)

[3]  Institute of Evolution, University of Haifa, Mount Carmel, 199 Abba Khoushi Ave., Haifa 3498838, Israel

[4]  Department for the Study of Mammoth Fauna, Academy of Sciences of the Republic of Sakha (Yakutia), 677007 Yakutsk, Russia; pavlovinn@mail.ru

*  Correspondence: sophia@evo.haifa.ac.il; Tel.: +972-4824-97-99

**Abstract:** In the context of global climate change, a significant increase in the active layer thickness (ALT) of permafrost is expected in the current century. This process has been observed by researchers over the past few decades. If the current climate trend continues, an increase in ALT may have a significant impact on the concentration of organic matter in Arctic river waters. The relationship between the thickness of the seasonally thawed layer of permafrost and the concentration of dissolved organic matter in river waters has been explored using clustering, one-way ANOVA, and cross-tabulation analysis. The data set for analysis included original details on the content of organic matter in the rivers of Eastern Siberia (in terms of COD, $BOD_5$, and the color of the water), phytoplankton abundance and biomass, and data on the permafrost active layer thickness (ALT) in the catchments. It was revealed that in the areas of catchments where the ALT is deeper, the content of organic matter in the rivers is lower than in areas with a shallow, seasonally thawed permafrost. Our results are consistent with the existing conceptual model of the influence of ALT on the chemistry of river waters in the cryolithozone. This knowledge is important for predicting the chemical composition of the Arctic rivers, eutrophication, and the rate of inflow of dissolved solids into the Arctic Ocean under the current conditions of ALT deepening.

**Keywords:** organic matter; COD; color of water; phytoplankton; active layer; permafrost; large rivers; Eastern Siberia

## 1. Introduction

In the soils of the Arctic catchments, due to swamping and the predominance of anaerobic conditions, extensive reserves of organic matter have been accumulated [1–3]. The upper soil horizons, in which organic matter is deposited, are the key source of organic matter in the river waters of the region [4]. The influx of organic matter into river waters is associated with the stimulation of the vegetation of planktonic algae, and, consequently, with eutrophication [5,6]. The intensity of vegetation of algae as autotrophic organisms is mainly regulated by the content of inorganic nutrients in the water, such as nitrogen and phosphorus compounds. However, some taxonomic groups of algae that are mixotrophs are capable of direct consumption of dissolved organic matter. As mixotrophic algae are usually scarce in Arctic rivers, phytoplankton and even autotrophic algae often utilize nutrients of organic form, so the mixotrophs will be omitted in the data of abundance and biomass. Therefore, variables of water quality such as oxygen consumption for chemical and biological processes (chemical oxygen demand (COD) and biochemical oxygen demand (BOD)) are proportionally related to phytoplankton biomass [7]. The natural process of

the transfer of dissolved organic matter from soils to river waters occurs due to surface runoff from the catchment. Under permafrost conditions, this process can be affected by the thickness of the seasonally thawed permafrost layer, or the so-called active layer thickness (hereinafter, ALT) [8]. There is uncertainty regarding how the ALT depth controls the input of organic matter into Arctic river waters. There is evidence that meltwater from permafrost soils formed during the seasonal degradation of permafrost leads to an increase in the removal of organic compounds [9]. A comparison of data on river basins with discontinuous and sporadic permafrost distribution showed that the catchments of Alaska [8,10–12], the Yukon Territory in Canada [13], and Central Siberia [14,15] are characterized by a reduced concentration of dissolved organic compounds in waters of rivers flowing through territories devoid of permafrost, where surface runoff is not blocked within the ALT. The waters of the tributaries of the Lena River, flowing in areas with a deep ALT, also found a lower carbon content, in comparison with tributaries located in areas with a shallow ALT [16]. However, observations in the catchments of the West Siberian Lowland show a low content of organic matter in the rivers of the permafrost zone, and a significantly higher content in areas outside the permafrost zone [17]. In the territory of Russia, the ALT increased by an average of 20 cm over the period of 1956–1990, according to Frauenfeld et al. [18]. Modeling of the ALT dynamics showed the possibility of its increase by 30–40% for most permafrost areas in the Northern Hemisphere by the year 2100 [19]. Despite the fact that forecasts of the state of permafrost in a changing climate have some variability [20], in the coming decades, an increase in the depth of the ALT is inevitable under the influence of the expected continued increase in air temperature [21]. In this regard, the issues of studying the effect of ALT on the formation of the chemical composition of surface waters become more relevant, since this knowledge is important for predicting possible changes in the chemistry of river waters in the Arctic and the rate of entry of dissolved substances into the Arctic Ocean [22]. The significance of such studies in the catchments of Eastern Siberia is even higher, given that in areas of continuous permafrost, to which this region belongs, transformations caused by permafrost degradation can be the most dramatic [23].

The aim of this work is to study the specific features of the influence of the ALT in the catchments of Eastern Siberia on the content of organic matter in river waters and phytoplankton.

## 2. Materials and Methods

The materials used for the study were the collected samples from the 12 largest rivers of Eastern Siberia: Lena, Vilyuy, Kolyma, Aldan, Olenyok, Vitim, Indigirka, Amga, Olyokma, Anabar, Yana, and Chara. Observations were carried out from 2007–2011 from June to August. Water samples were taken from the surface horizon (0–0.3 m) both in the coastal areas and along the fairways of the rivers. Preservation and storage of the water samples for subsequent chemical-analytical processing in the laboratory was carried out in accordance with the guidelines for the chemical analysis of land surface waters [24]. Three variables were used as indicators of the content of organic matter in water; COD, $BOD_5$, and water color. Determination of the $BOD_5$ value was carried out at the sampling site by a titrimetric method based on iodometric determination. The remaining indicators were determined in the laboratory of the Institute for Biological Problems of Cryolithozone of the Siberian Branch of the Russian Academy of Science by the photometric method: COD-on the device "Fluorat-02", Lumex, Russia. Water color was measured on the spectrophotometer SF-26 (LOMO, Saint Petersburg, Russia) according to the platinum-cobalt (Pt-Co) scale ranges from 0 to 500 units. Information on the content of organic matter, phytoplankton abundance, and biomass for each observation point was published by us previously [25].

A correlation analysis between the group "organic matter" environmental variables as independent variables, and the phytoplankton abundance and biomass as dependent variables, was carried out based on the Pearson coefficients calculation in the ExStatR Program ver. 1.2 [26].

Data on the ALT of permafrost were obtained on the basis of materials published by C. Beer et al. [27], as well as on the basis of the map of Russia "Seasonal freezing and thawing of soils" [28].

The data array we formed included eight quantitative variables, which were grouped into three groups: organic matter, characteristics of the seasonally thawed permafrost layer, and geographical coordinates of sampling points (observations) (Table 1). The total number of observations of the array is 303.

**Table 1.** Variables of the analyzed data array.

| Variable Groups | Variable Name |
|---|---|
| Organic matter | Color, Pt-Co units<br>COD, mg L$^{-1}$<br>BOD$_5$, mg L$^{-1}$ |
| Characteristics of the seasonally thawed permafrost layer | Minimum Active Layer Thickness, m<br>Maximum Active Layer Thickness, m<br>Mean Active Layer Thickness, m |
| Geographical coordinates of sampling points (observations) | East, degrees<br>North, degrees |

For clustering, the Euclidean distance was utilized using the Ward algorithm [29]. Since the analyzed features had different scales, in order to eliminate the dominance of individual features with maximum numerical values, a standardization procedure was preliminarily carried out. Differences between clusters were assessed by comparing the group means for array variables using one-way ANOVA [30].

The cross-tabulation analysis of the gradations of the qualitative grouping variables was performed using Pearson's chi-square ($\chi^2$) goodness-of-fit test, and Cramer's V-criterion was used as an indicator of the intensity of the relationship [31–33]. Threshold values for the Cramer V-criterion were taken in accordance with A. M. Grzhibovsky [34]. The use of this approach, and the application of the actual data obtained during the research, in the absence of theoretical frequencies (obtained from the theoretical distribution law), involved the artificial construction of some theoretical frequencies with which the comparison was made. Theoretical frequencies were calculated on the basis of actual data, based on the condition of independence of two features. Using the chi-square test, the actual (observed) and calculated, and theoretical (expected) frequencies were compared. Moreover, if the actual frequency was higher than the theoretical one, then the relationship (as well as the difference between these frequencies) was positive, and vice versa.

When testing statistical hypotheses, the critical level of statistical significance was assumed to be 5%. Statistical analysis procedures were performed using the Statistica 10 software package.

## 3. Results

Using the clustering of observations of the data array by variables from the group "organic matter", the grouping variable Cl_2O "indicators of the content of organic matter in water" was obtained. The results of comparing the means of variables for two selected clusters using one-way ANOVA indicated the maximum difference between the gradations of the new categorical variable Cl_2O, in terms of water color and COD (Table 2).

The plot of the standardized means for each cluster of Cl_2O (Figure 1) shows that the two resulting clusters clearly differ in all five analyzed variables. The first cluster was characterized by higher mean values of the variables.

**Table 2.** Analysis of variance and means for each cluster of Cl_2O.

| Variable | Clusters | | F | p |
| --- | --- | --- | --- | --- |
| | 1 | 2 | | |
| Color, Pt-Co units | 58.27 | 16.98 | 577.6 | 0.0000 |
| COD, mg L$^{-1}$ | 41.82 | 21.99 | 184.4 | 0.0000 |
| BOD$_5$, mg L$^{-1}$ | 1.11 | 0.86 | 9.2 | 0.0027 |

Note. The following designations are used in the header of the table: *F*—Fisher's F-test, *p*—significance level.

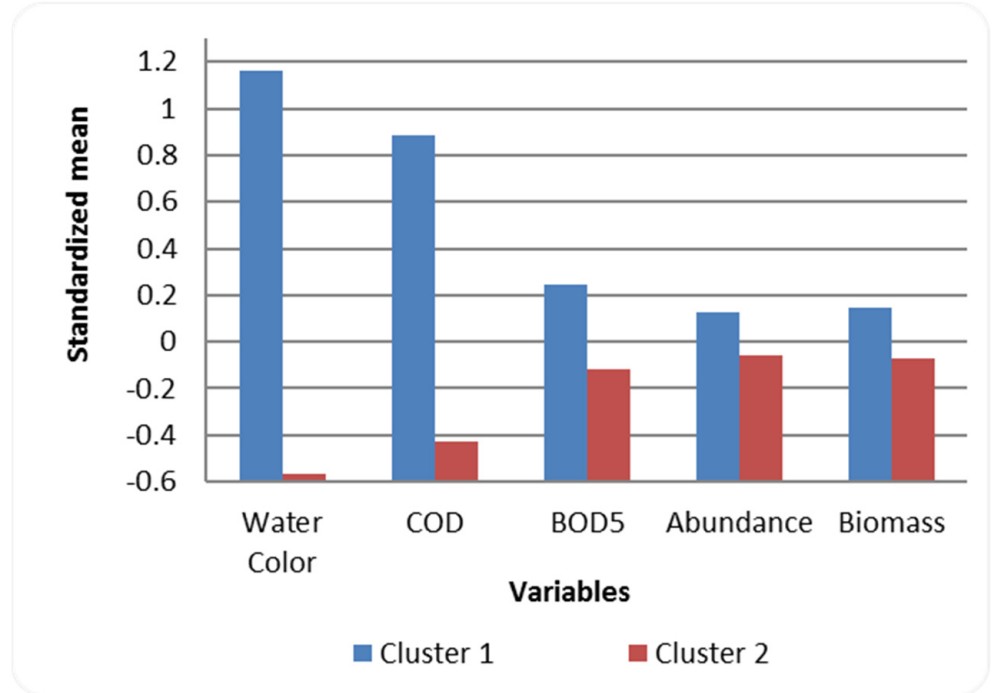

**Figure 1.** Plot of standardized means for each cluster of Cl_2O.

We conducted a correlation analysis of the studied variables based on the Pearson coefficients. The data are shown in Table 3. It can be seen that the watercolor and phytoplankton abundance are the most related. COD and BOD are also related, and the abundance and biomass of phytoplankton are significantly correlated with BOD$_5$.

**Table 3.** Correlation analysis of variables from the group "organic matter" and phytoplankton abundance and biomass, according to [25], based on Pearson's correlation coefficient with significance values * $p < 0.05$.

| Variable | Water Color | COD | BOD$_5$ | Abundance | Biomass |
| --- | --- | --- | --- | --- | --- |
| Water Color | 0 | 0.48 | 0.15 | **0 *** | 0.08 |
| COD | 0 | 0 | **0.02 *** | 0.08 | 0.19 |
| BOD$_5$ | 0.009 | **0.783** | 0 | **0.05 *** | **0.04 *** |
| Abundance | **0.967** | 0.177 | **0.423** | 0 | 0.36 |
| Biomass | 0.190 | 0.001 | **0.442** | 0 | 0 |

Note: the left lower part are the Pearson coefficients, the upper right part represents the *p*-value. Significant values are in bold.

The identified clusters were highly geographically localized (Figure 2). The first cluster, richest in organic matter, combined observations on the rivers of the northwestern part of the studied region (Anabar, Olenyok, and Vilyuy), as well as individual observations on the Indigirka, Kolyma, Amga, Aldan, and Olyokma rivers. Observations of the second cluster were concentrated mainly along the rivers of the northeast and south of the region.

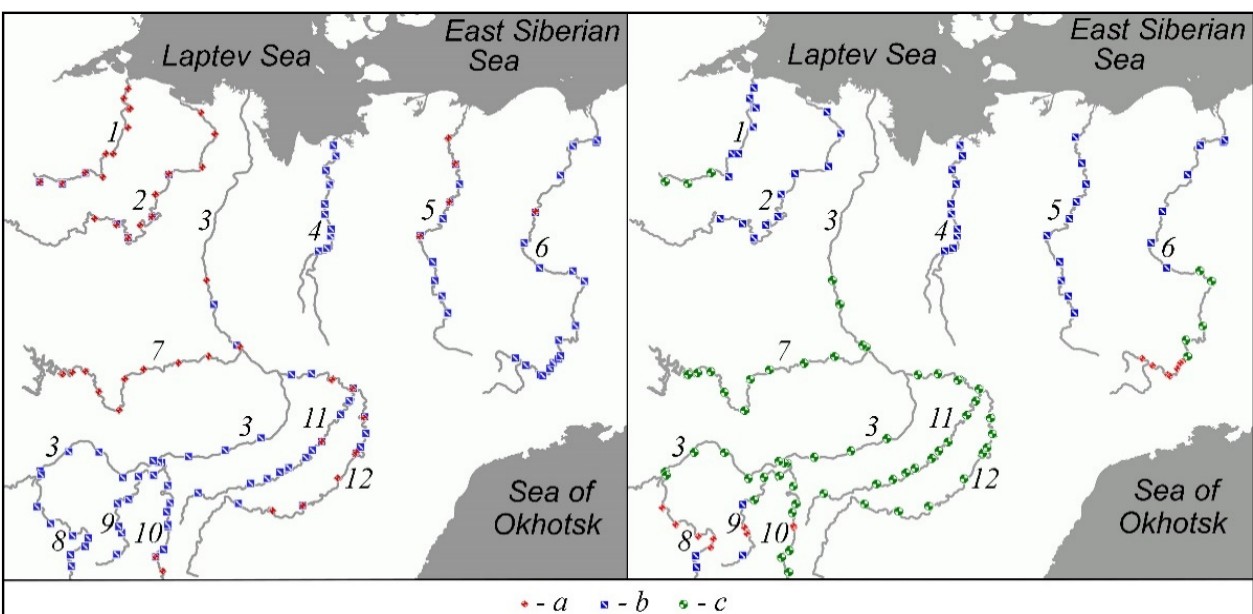

**Figure 2.** Observations classified according to the Cl_2O gradation (**left panel**) and according to the Cl_3P gradation (**right panel**) (a—cluster 1, red; b—cluster 2, blue; c—cluster 3, green; rivers are marked with numbers: 1—Anabar, 2—Olenyok, 3—Lena, 4—Yana, 5—Indigirka, 6—Kolyma, 7—Vilyuy, 8—Vitim, 9—Chara, 10—Olyokma, 11—Amga, 12—Aldan).

In addition, observations were clustered according to indicators from the group "Characteristics of the seasonally thawed permafrost layer" (hereinafter, gradation Cl_3P "properties of permafrost"). According to the ranked values of the F-criterion, the greatest difference between the clusters was achieved in terms of the minimum and maximum ALT (Table 4).

**Table 4.** Analysis of variance and means for each cluster of Cl_3P.

| Variable | Clusters | | | F | p |
|---|---|---|---|---|---|
| | 1 | 2 | 3 | | |
| Minimum Active Layer Thickness, m | 0.96 | 0.19 | 0.46 | 916.1 | 0.0000 |
| Maximum Active Layer Thickness, m | 3.24 | 1.58 | 3.51 | 800.5 | 0.0000 |
| Mean Active Layer Thickness, m | 2.00 | 0.95 | 1.78 | 302.0 | 0.0000 |

Note. The following designations are used in the header of the table: *F*—Fisher's F-test, *p*—significance level.

An analysis of the standardized means for each cluster of Cl_3P showed that the second cluster differed from the others in the lowest degree of seasonal thawing of permafrost for all three measured variables (Figure 3). All three clusters were well distinguished by the variable "Minimum Active Layer Thickness".

The second cluster, characterized by the lowest degree of seasonal permafrost thawing, combined observations on Arctic rivers (Figure 2), including them in either their entirety (Olenyok, Yana, Indigirka) or their lower sections (Anabar, Kolyma), as well as individual observations from the upper-reaches of mountain rivers in the south of the region (Vitim, Chara). The first cluster, whose observations were characterized by the highest minimum thickness of seasonal thawing of permafrost, was localized in the south of the region in sections of the Kolyma, Vitim, Chara, and Olyokma rivers.

The third cluster, which occupied an intermediate position among the three identified gradations Cl_3P, combined observations of the central part of the studied region, including the Vilyuy, Lena, Aldan, and Amga rivers, and some sections of the Anabar, Kolyma, Olyokma, and Chara rivers.

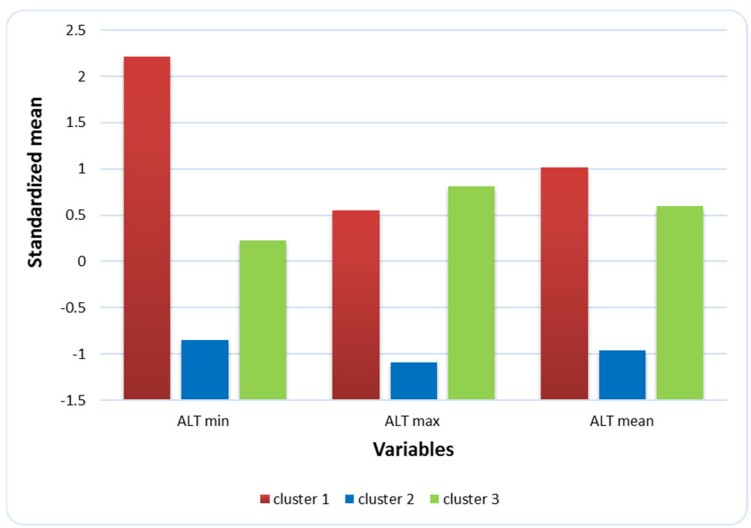

**Figure 3.** Plot of standardized means for each cluster of Cl_3P.

A cross-tabulation analysis of the categorical variables Cl_2O and Cl_3P was performed to find the relationship between the ALT in the catchment and the concentration of organic matter in river waters. The value of Cramer's V-test (0.25) indicated that the intensity of the relationship between these two variables is medium, and the level of significance achieved ($p = 0.0001$) confirmed the suitability of the data for further analysis. The value of the chi-square test, which is the sum of the contributions by all cells of the pivot (Table 5) is 18.29. The maximum values of the chi-square test, and, consequently, the strength of the relationship between the features, correspond to the cells at the intersection of the row "cluster 1" of the categorical variable Cl_3P and the columns "cluster 1" and "cluster 2" of the gradation Cl_2O (Table 5). The differences between the observed and expected frequencies for these cells of the pivot Table 5 were significant. Thus, the first cluster of gradation Cl_3P, which was characterized by the highest degree of seasonal permafrost thawing, has a negative relationship with the first, most organic-rich cluster of the categorical variable Cl_2O, and vice versa, a positive relationship with the second cluster of waters poor in organic matter.

**Table 5.** Crosstabulation results of qualitative variables Cl_2O "indicators of the content of organic matter in water" and Cl_3P "properties of permafrost".

| Summary Results | | | Cl_2O | | All Gradations |
|---|---|---|---|---|---|
| | | | Cluster 1 | Cluster 2 | |
| Observed frequency | | Cluster 1 | 0 | 33 | 33 |
| Expected frequency | | | 10.78 | 22.22 | - |
| Observed minus expected frequencies | | | −10.78 | 10.78 | 0 |
| Chi-square by cell | | | 10.782 | 5.233 | - |
| Observed frequency | Cl_3P | Cluster 2 | 48 | 77 | 125 |
| Expected frequency | | | 40.84 | 84.16 | - |
| Observed minus expected frequencies | | | 7.16 | −7.16 | 0 |
| Chi-square by cell | | | 1.255 | 0.609 | - |
| Observed frequency | | Cluster 3 | 51 | 94 | 145 |
| Expected frequency | | | 47.38 | 97.62 | - |
| Observed minus expected frequencies | | | 3.62 | −3.62 | 0 |
| Chi-square by cell | | | 0.277 | 0.135 | - |
| Observed frequency | All gradations | | 99 | 204 | 303 |

## 4. Discussion

Arctic rivers are characterized by a high content of organic matter [35], which is due to the influence of carbon-rich soils in catchments located in the permafrost zone. Our study has shown the presence of a gradient in the content of organic matter in the waters of the rivers of Eastern Siberia (Table 2, Figure 2). A comparison of our data of oxygen demand and the abundance and biomass of phytoplankton demonstrated a high correlation of these variables in the Arctic rivers (Table 3). As can be seen from the data of Gibson et al. [27], the production of organic matter in the zone of water removal by the rivers of Eastern Siberia, as well as the Yenisei and Ob rivers, is the highest for the entire Arctic Ocean. This emphasizes the importance of the present study of the sources of organic matter exports to the Arctic Ocean.

The ALT, despite its more complex distribution in mountainous regions and in zones of discontinuous permafrost in the south of the region, is generally characterized by an increase in the direction of north to south [27]. This is also shown by the result of our analysis (Table 4, Figure 2). A comparison of patterns of spatial distribution of ALT in the catchments of the studied region, and the content of organic matter in river waters, revealed the following pattern: the deeper the layer of seasonal thawing of permafrost in the catchment area, the lower the concentration of organic matter in river waters.

The revealed patterns are explained by the differences in the depths of the prevailing runoff routes from catchments, which, in turn, is controlled by the thickness of the seasonally thawed permafrost layer. Our result generally agrees with the main provisions of the conceptual model of the effect of ALT on the chemical composition of inner waters, which was developed by R. MacLean et al. [8] for the catchments of Alaska. In accordance with the conceptual model, permafrost retains runoff in the upper layer of soil that is rich in organic matter, where surface runoff waters are saturated with dissolved organic matter. As a result, this leads to an increase in the content of organic matter in river waters. In catchments with a deeper ALT, the underlying mineral soil layer becomes part of the seasonally thawed layer. Thus, surface runoff provides access to the mineral soil horizon, contact in which surface runoff waters lose dissolved organic matter due to the process of abiotic adsorption [36].

A number of researchers obtained similar results by comparing data on the concentration of organic matter in rivers flowing through areas of catchments devoid of permafrost and across the permafrost zone. Such studies were carried out in areas of sporadic and discontinuous distribution of permafrost in the river basins of Alaska [8–12], the Yukon Territory in Canada [13], and Central Siberia [14,15]. However, similar studies in the catchments of Western Siberia led to opposite results: in areas outside of the permafrost zone, where permafrost did not block surface runoff water in the redistribution of the surface soil horizon, the researchers noted not a decrease, but an increase in the concentration of organic matter in river waters [17]. The researchers explained this result by the presence of a deep layer of peatlands in the West Siberian Lowland, which has an average depth of 1–5 m [37], and provides an additional source of organic matter in areas of catchments devoid of permafrost, where surface runoff is not blocked within the ALT. Thus, an increase in the ALT in Western Siberia opens up access to the lower horizons of peat soils that are even richer in organic matter, and contributes to an increase in the export of organic matter to the surface waters of the region.

## 5. Conclusions

Obviously, in the area of permafrost, a factor such as the ALT has a great potential for influencing the chemical composition of surface waters. In the drainage basins of the permafrost zone, vast reserves of organic matter have been accumulated, which are recorded in the upper horizon of permafrost soils. The predominant routes of surface runoff from catchments, determined by the ALT, are able to regulate the natural process of removal of dissolved organic matter from soils. The increase in ALT leads to a change

in the depth of surface runoff and the duration of water stay in the soil, which affects the chemical composition of inner waters through exchange reactions between water and soil.

In this study, we used original data on the content of organic matter and phytoplankton in the waters of the largest rivers of Eastern Siberia, as well as published information on their ALTs. As a result of the analysis, it was found that such indirect indicators of the content of organic matter in water, such as COD and water color, and (indirectly) the abundance and biomass of phytoplankton, were correlated and associated with the degree of seasonal permafrost degradation. It was shown that in catchments with a less thick seasonally thawed layer of permafrost, the content of organic matter in river waters is higher than in those parts of the studied region where the degree of seasonal degradation of permafrost is greater. This is consistent with the elements of the existing concept of the regulatory role of the ALT in the formation of the chemical composition of permafrost waters, which affects eutrophication.

An increase in the ALT in recent decades has been observed for the entire territory of the permafrost zone of the northern hemisphere, and there are forecasts of its increase in the current century. Therefore, the results obtained are important from the point of view of assessing possible changes in the chemistry of river waters in the Arctic in the future and the rate of entry of dissolved substances into the Arctic Ocean.

**Author Contributions:** Conceptualization, O.I.G. and V.A.G.; methodology, O.I.G.; software, I.A.Y.; validation, O.I.G., V.A.G., S.B. and I.A.Y.; formal analysis, I.A.Y.; investigation, O.I.G.; resources, V.A.G.; data curation, S.B.; writing—original draft preparation, O.I.G., V.A.G. and S.B.; writing—review and editing, S.B.; visualization, I.S.P.; supervision, S.B.; project administration, V.A.G.; funding acquisition, I.A.Y. All authors have read and agreed to the published version of the manuscript.

**Funding:** The research was carried out within the state assignment of the Ministry of Natural Resources and Environment of the Russian Federation (theme No. 1-22-81-4).

**Institutional Review Board Statement:** Not applicable.

**Informed Consent Statement:** Not applicable.

**Data Availability Statement:** Not applicable.

**Conflicts of Interest:** The authors declare no conflict of interest.

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
