# Peer review of "Influence of the Thickness of the Seasonally Thawed Layer of Permafrost in the Eastern Siberia Catchments on the Content of Organic Matter in River Waters"

_2673-9917, doi:10.3390/hydrobiology1020018_

Round 1

Reviewer 1 Report

This paper aims to propose a statistical model to explain the changes in organic matters in Eastern Siberia rivers. The work is very well written with very minor issues showed in the attachment. However, there are two major issues to be addressed by the authors:  

- The background of the study justifying the rationale of the study is not presented in the abstract. Even if the Introduction, this needs to be strengthened.

- I do not see the statistical model the authors are talking about. I only saw a one-way ANOVA and some correlation tests. The methodology therefore needs to be revised in such a way that the model fitted is clearly presented. The statistical formula of the model be presented and the predictors and response variables be clearly indicated. 

Author Response

Thank you the Reviewer 1 for comments that improved my ms. Please find below the responses to each comments

With best regards,

Prof Sophia Barinova

Reviewer: The background of the study justifying the rationale of the study is not presented in the abstract. Even if the Introduction, this needs to be strengthened.

Authors: added

Reviewer: I do not see the statistical model the authors are talking about. I only saw a one-way ANOVA and some correlation tests. The methodology therefore needs to be revised in such a way that the model fitted is clearly presented. The statistical formula of the model be presented and the predictors and response variables be clearly indicated.

Authors: Rewritten, explanation added to correlation analysis.

Reviewer 2 Report

Dear Authors,

After reading the submitted manuscript I have pointed out some issues that need to be clarified before the article can be published. Please consider the below comments.

Line 37: I have a general comment about the role of organic compounds regarding eutrophication development. As your study focuses on organic compounds I would add that for many species of aquatic vegetation globally inorganic nutrient compounds are much more critical due to their higher availability. For some species, the organic compounds can of course be responsible for limiting eutrophication.

I would also consider deleting the "and their eutrophication" from the title of this paper while eutrophication is a very complex process and not only organic matter from rivers are responsible for its dynamics.

Line 75: The observations were carried out more than 10 years ago so I would expect a sentence of explanation why were the Authors waiting so long with this publication.

Line 85: What is the range of "colours" given by the device SF-26. Please add that SF-26 is a spectrophotometer 

Table 1: Explain what "degree" means

Line 247: In the conclusions please add key results according to the Guide for Authors

Author Response

Thank you the Reviewer 2 for comments that improved my ms. Please find below the responses to each comments

With best regards,

Prof Sophia Barinova

Reviewer: Line 37: I have a general comment about the role of organic compounds regarding eutrophication development. As your study focuses on organic compounds I would add that for many species of aquatic vegetation globally inorganic nutrient compounds are much more critical due to their higher availability. For some species, the organic compounds can of course be responsible for limiting eutrophication.

Authors: added

Reviewer: I would also consider deleting the "and their eutrophication" from the title of this paper while eutrophication is a very complex process and not only organic matter from rivers are responsible for its dynamics.

Authors: agree, deleted

Reviewer: Line 75: The observations were carried out more than 10 years ago so I would expect a sentence of explanation why were the Authors waiting so long with this publication.

Authors: The data were obtained earlier, but new data are unlikely to be obtained again in the near future due to the fact that the territory is very vast and extremely difficult for researchers to access. At the same time, climatic processes do not proceed quickly, which makes it possible to use data from previous observations. This allows us to be sure that the relevance of these data is still not lost and their new interpretation will be appropriate in this study.

Reviewer: Line 85: What is the range of "colours" given by the device SF-26. Please add that SF-26 is a spectrophotometer 

Table 1: Explain what "degree" means

Authors: This misunderstanding was our fault. We have made appropriate changes to the text. Here we explain that we used a conventional method that based on Pt/Co scale. It ranges from 0 to 500 units. In tables 1 and 2, "degree" was replaced with "units".

Reviewer: Line 247: In the conclusions please add key results according to the Guide for Authors

Authors: The conclusion included and rather briefly presents all the results obtained by us as well as the conclusions that followed them.

Round 2

Reviewer 1 Report

About my earlier comments on the method, the authors wrote:

"Correlation analysis between the group "organic matter" environmental variables as independent and the phytoplankton abundance and biomass as dependent variables was doing on the base of Pearson coefficients calculation in ExStatR Program ver. 1.2 [26]".

This statement does not read well at all. Here are some points of issues"

- In a correlation analysis, there is no independent and dependent variables, unless you fit a model to your data, which is not the case here since you run only a Pearson correlation test. How do you know which variables are dependent/independent in a correlation analysis?

- "...organic matter" environmental variables". Open a bracket after "variables to list few of those environmental variables you are referring to.

- English: You wrote "....dependent variables was doing...". It cannot be "was doing" but "was done"

Lines 149-150: "Table 3. Correlation analysis of variables from the group "organic matter" and phytoplankton abundance and biomass according [25] based on Pearson's correlation coefficient with Significance values* p < 0.05" 

- Not "according [25]" but "according to [25].

I suggest the authors go through the manuscript with the help of an english editor if need be to make sure the language mistakes/spelling are corrected.

Author Response

(The authors gave the same response as above.)
